# Interrelationship between the Microbial Communities of the Root Canals and Periodontal Pockets in Combined Endodontic-Periodontal Diseases

**DOI:** 10.3390/microorganisms9091925

**Published:** 2021-09-10

**Authors:** Erica M. Lopes, Maicon R. Z. Passini, Luciano T. Kishi, Tsute Chen, Bruce J. Paster, Brenda P. F. A. Gomes

**Affiliations:** 1Department of Restorative Dentistry, Division of Endodontics, Piracicaba Dental School, State University of Campinas -UNICAMP, Piracicaba 13400-001, SP, Brazil; maicon_plis@hotmail.com; 2National Laboratory of Scientific Computing, Petrópolis 25715-183, RJ, Brazil; luciano.kishi@gmail.com; 3Department of Molecular Genetics, The Forsyth Institute, Cambridge, MA 02142, USA; tchen@forsyth.org; 4Microbiology Department, The Forsyth Institute, Cambridge, MA 02142, USA; bpaster@forsyth.org

**Keywords:** microbiome, metagenomics, endodontics

## Abstract

Periodontal and Endodontic diseases are biofilm-related diseases. The presence of microorganisms in root canals (RCs) and the complex microbiota of periodontal pockets (PPs) contribute to the development of endodontic-periodontal diseases. This study performed a systemic analysis using state-of-the-art sequence data to assess the microbial composition of infected RCs and PPs to further assess the microbiota and verify the possibility of cross-infection between these sites. The microbiomes of these combined diseases were examined with a focus on the V3-V4 hypervariable region of the 16S rRNA gene. The number of species in PP was higher than in RC, and there was a predominance of obligate anaerobes and gram-negative bacteria. In the RCs, the genera *Enterococcus*, *Parvimonas*, *Stomatobaculum* predominated, in contrast, the PPs revealed a predominance of *Enterococcus*, *Parvimonas*, *Stomatobaculum*, *Peptostreptococcus* and *Mogibacterium.* The RC and PP microbiome was not similar with regards to the sharing of OTUs for phyla and genera (8 and 67, respectively). The evaluation of molecular markers revealed a large number of markers for resistance to antibiotics of the carbapenem and beta-lactam type (broad spectrum). Another relevant finding of this study was the markers related to systemic diseases related to cardiac muscle and rheumatology, among others. In conclusion, the RC microbiota was less complex and diverse than PP. Interactions between microbial communities were present. The shared genus can signal communication between the endodontic and periodontal microbiomes.

## 1. Introduction

Endodontic-periodontal diseases consist of pulp and periodontal disease on the same tooth [1].

Infection in pulp tissue, if not treated, progresses to pulpal necrosis and frequently leads to periodontal tissue breakdown (apical periodontitis). In contrast, severe periodontal disease may initiate or exacerbate the inflammatory changes in the pulp tissue. However, a lesion in the pulp occurs when periodontal disease advances from the apex of the root to the main foramen [1,2,3,4,5]. Lesions arising from combined periodontal endodontic diseases are a great challenge for professionals in the field of endodontics/periodontics in terms of diagnosis and prognosis of the teeth involved. A series of etiological factors such as bacteria, fungi, viruses and promoting factors such as trauma, root resorption, perforations and dental malformations are protagonists in the advancement and progression of these lesion [2,3]. The prognosis and treatment of endo-perio lesions vary according to the diagnosis and condition of the patient. However, in cases of combined endodontic-periodontal diseases, there is a need for combined therapy, that is, root canal (RC) and periodontal treatments. [6].

The most frequent pathologies found in the oral cavity are caused by microbial biofilms formed and adhered to the surface of the teeth, such as dental caries and periodontitis. [7,8,9]. Endodontic and periodontal diseases are both biofilm-derived diseases. Biofilms are defined as sessile microbial communities that adhere to hard surfaces. The microorganisms that make up the biofilm form an extremely organized community, surrounded by an extracellular matrix composed mainly of polysaccharides produced by the microorganisms themselves, which interact with components of the fluid in which they are bathed. Biofilms are typically bathed in fluids that carry microorganisms. Dental biofilm is bathed in the saliva.

Many microorganisms from the most varied natural ecosystems form biofilm matrices. Their development in injured and affected tissues is affected by communities closed by a matrix [7,10,11,12].

Biofilms generally comprise 15% of the cells incorporated (by volume) into 85% of the matrix material. This complex structure is divided into ramifications of water ducts that transport fluid mass to the community via connective flow [13] of eukaryotic cells [13,14].

The biofilm formation process follows a developmental sequence, a mature community formed by tower- and mushroom-shaped micro colonies, with variations according to the predominant species of species in the community. Sequence events follow the following order, which is the correction of the microbial surface, cell proliferation, production and detachment of the matrix [15]. Biofilm dormancy, adhesion and deadhesion are controlled through chemical signals, which guide the structuring of limited and closed biofilm micro colonies with exopolysaccharides and water [14]. Biofilm formation by microorganisms is a defense and survival process [16]. In environments with extreme conditions such as high temperatures, acidity, safety, they promote the inherently defensive dormancy process because bacterial cells are not transported to lethal areas [14].

Infectious biofilms, with a high degree of pathogenicity, are difficult to detect by routine diagnosis and are inherently tolerant as host-recurring antibiotic therapies [15]. Furthermore, biofilms are spreaders of antibiotic resistance strands, from the horizontal gene transfer process. The microorganisms present in this matrix are extremely adapted to adverse conditions such as: environmental stresses, such as changes in nutritional quality, cell density, temperature, pH and osmolality [17]. The low rate of detection of biofilm infection is in the laboratory methods used for its identification. So far, the role played by biofilms in the interrelationship between endodontic and periodontal diseases has not been fully elucidated.

Considering the relationship of microorganisms as etiological factors in endodontic-periodontal diseases and that the inter-microbial communication between these sites is related to disease progression, this study conducted an in-silico study with NGS data to assess the microbial composition of the infected RCs and periodontal pockets (PPs) and to verify the possibility of cross-infection between these sites.

## 2. Materials and Methods

### 2.1. Study Population

Twelve patients with combined endodontic-periodontal diseases were selected for this study. They were screened and indicated by the School of Dentistry of Piracicaba, State University of Campinas, UNICAMP, Piracicaba, SP, Brazil at the time they were seeking dental treatment. The clinical examination revealed that the pulp was necrotic and the clinical signs of periodontal disease presented the lesion parameters described by Gomes et.al. (2015). 

Pulp status was assessed using thermal vitality tests and a cavity test. The apical condition was evaluated based on the presence or absence of clinical signs or symptoms during the percussion and palpation tests. Periodontal conditions were assessed using a periodontal probe at six sites on the affected tooth and radiographs.

This study was approved by the Ethics Committee on Research in Human Beings, Faculty of Dentistry of Piracicaba, State University of Campinas (UNICAMP), Piracicaba (SP), Brazil, according to protocol number CAAE 86140218.0.0000.5418 approved by the Municipal Health Council - Ministry da Saúde on 10 August 2018. All patients signed an informed consent form prior to their participation and the register of cases included in this study can be found in the Brazilian Registry of Clinical Trials (ReBEC; UTN U1111-1238- 5402). 

### 2.2. Sample Collection

Periodontal pocket and root canal samples were collected before starting both treatments. The samples were collected following the protocol established by Gomes et al. (2015). Three consecutive sterile paper points were kept in place for 60s, and then added into a sterile tube containing 1 mL of VMGA III transport medium, followed by freezing at −70 °C until processing.

### 2.3. Evaluation of the RC and PP Microbiomes

Research to assess the interrelationship (communication) of the microbiota of RCs and PPs was conducted using a metagenomic approach and next-generation sequencing. DNA was extracted from 12 RCs and PPs using the QI-Aamp DNA Micro kit (QIAGEN, Valencia, CA, USA). Microbiomes were examined focusing on the V3-V4 hypervariable region of the 16S rRNA gene from Illumina MiSeq sequencing (Illumina, San Diego, CA, USA) by the Forsyth Institute (Cambrige, MA, USA) using a microbe identification modification oral human using next-generation sequencing protocol, according to Gomes et al. [2] and Mougeot et al. [18].

The raw data from the sequencing were punched for pre-processing and removing adapters and strings with Phred quality below 20 using the BBduk program from the BBmap package (https://github.com/BioInfoTools/BBMap; accessed on 10 September 2020). The QIIME version 1.9.1 package was used to filter reads and determine operational taxonomic units (OTUs) [19]. The microbiota at both sites was investigated using an internal pipeline, based on QIIME [19], which performs all pre-processing, and comparison with HOMID [20], RDPII [21], Greengenes [22] and EZBiocloud databases [23]. A curatorial process was performed to remove OTUs from repeated taxa, so that there was no annotation in duplication or overlapping of the taxa.

Statistical and comparative analyses were performed using the Microbiome Analyst resource [24]. The reads submitted for analysis were filtered according to the following parameters: (i) minimum count of four; (ii) count filter with a 10% prevalence. For comparative studies, a low-variance filter was applied, based on the interquartile range and characteristics below 10% were removed. Data were normalized considering the minimum library size and normalized based on the total value of reads obtained before any statistics comparison [24]. The statistical studies performed included Venn diagram, clustering, correlation, co-occurrence, and calculation of diversity indices between observed OTUs for phylum, genus, and species.

## 3. Results

### 3.1. Microbiome of the RCs and PPs

Illumina sequencing of 16S rRNA V3-V4 region amplicon libraries generated from RCs and PPs (*n* = 12) generated 1,792,885 raw reading pairs, of which 568,522 passed the quality control steps. After clustering the OTUs, the data were normalized to 39,544 reads.

Microbiomes of the RCs showed the presence of eight bacterial phyla, including *Firmicutes* (75.68%), *Proteobacteria* (10.5%), *Actinobacteria* (7.96%), *Bacteroidetes* (4.03%), *Synergistetes* (1.43%), *Spirochaetes* (0.2%), *Chloroflexi* (0.14%), and *Fusobacteria* (0.07%) (Figure 1a).

A total of 68 bacterial genera were identified in the RCs, 37 of which had a percentage of OTUs above 0.1%. The RCs were predominant in *Enterococcus* (28.53%), *Parvimonas* (12.58%), *Stomatobaculum* (6.11%), *Peptoniphilaceae [G-1]* (5.32%), *Peptostreptococcus* (4.73%), *Mogibacterium* (3.64%), *Olsenella* (3.48%), *Bacteroidaceae [G-1]* (2.85%), *Filifactor* (2.41%), *Oribacterium* (2.41%), *Pseudomonas* (2.07%) and *Campylobacter* (1.66%) (Figure 1b).

The microbiomes of the PPs showed the presence of nine phyla, including *Firmicutes* (57.86%), *Proteobacteria* (24.8%), *Actinobacteria* (5.62%), *Saccharibacteria (TM7)* (3.72%), *Fusobacteria* (3.13%), *Bacteroidetes* (2.96%), *Synergistetes* (1.29%), *Spirochaetes* (0.39%) and *Chloroflexi* (0.23%) (Figure 1a).

A total of 73 bacterial genera were identified in the PPs, 47 of them with a percentage of OTUs above 0.1%, with the greatest predominance of *Streptococcus* (26.33%), *Desulfobulbus* (18.18%), *Enterococcus* (8.01%), *Parvimonas* (7.1%), *Saccharibacteria (TM7)* (3.72%), *Fusobacterium* (3.12%), *Peptoniphilaceae* (2.51%) and *Actinomyces* (2.45%) (Figure 1b).

Gram-negative and gram-positive bacteria are virulent factors that are related to the degree of infection. A total of 589,936 (RCs) and 278,145 (PPs) reads were annotated for gram-positive bacteria, while 81,358 (RCs) and 269,165 (PPs) reads were annotated for gram-negative bacteria (Figure 2). Picture 2 also shows other phenotypic characteristics of the taxa present in the RCs and PPs.

A study of beta microbiota diversity of RCs and PPs (Figure 3a) presented a similar overlap profile, with much less pronounced variations between comparisons. OTUs observed in RCs were much smaller than in PPs (Figure 3b). The Shannon index (Kruskal-Wallis test; *p* < 0.05) revealed a microbiota with much more pronounced values in PPs, which reveals the greater diversity of taxa found; it is noteworthy that the microbiota of both the sites was more diverse when considering the Shannon-Weaver values (Figure 3c). Species richness studies (Chao1) revealed a profile similar to that of Shannon’s studies, where PP cases stood out in relation to RCs (Figure 3d).

### 3.2. Comparative Studies of RC and PPs Microbiomes

The taxonomy of endodontic-periodontal diseases is illustrated using a heat tree (Figure 4). The nodes represent the abundance of taxan between samples and the branches represent the relationship between the study conditions. The richness of OTUs can be observed by the size of the node (thickness), as well as the intensity of color, branches and green nodes are the most predominant in RCs, in contrast, branches in yellow color represent greater abundance in PPs between the endodontic-periodontal compartments are driven by an increase in bacterial diversity in PPs.

The abundance of taxa corroborates the data described in Figure 2, where *Cardiobacterium*, *Streptococcus*, *Campylobacter*, *Pseudomomas*, among others, are represented by the line and nodes in the green color, which indicates greater abundance in CRs (Figure 3), whereas the PP microbiota has high prevalence of *Fusobacterium*, *Streptococcus*, *Capnocytophaga*, among others, visually represented in yellow lines and nodes. Figure 3 clearly represents that there are more genera identified in PPs than in RCs, as there are a larger number of yellow nodes than green nodes.

The Venn diagram of the phyla observed for RCs and PPs showed the presence of eight common taxa among the studied sites (Figure 5a). In contrast, a similar study conducted for genus revealed 68 common taxa between RCs and PPs. A total of five genera were identified only in PPs, including *Bacteroidales* [G-2], *Saccharibacteria* (TM7) [G-1], *Lachnoanaerobaculum*, *Aggregatibacter* and *Lachnospiraceae* G (Figure 5b).

The comparison between cases and conditions at the phylum level revealed three clusters (Figure 6a), and the Proteobacteria, Actinobacteria, and Firmicutes phyla showed a positive correlation owning to the higher prevalence in 14 of the 24 studies. Clustering between RCs and PPs at the genus level revealed 15 clusters, including Streptococcus, *Parvimonas*, *Prevotella*, *Actinomyces*, *Fusobacterium*, *Treponema* and *Filifactor* (Figure 6b).

Studies based on hierarchical grouping based on the taxonomy of the samples, revealed three clusters (Figure 7a). Some samples of the same study conditions were grouped, indicating similarity between the groups. It is speculated that the observed dispersion among some of the samples may be correlated with the similarity in the microbiota.

Relationships between the main abundant bacterial genera at the microbial community level were investigated using the Pearson’s correlation analysis. The microbiome of the RC and PP samples detected positive correlations (co-occurrence) for the genera *Helicobacter*, *Sulfurospirillum*, *Atopobium*, *Olsenella*, *Bosea*, *Kocuria*, *Hyphomicrobium*, *Rhodobacter*, *Paracoccus*, *Rhizobium*, *Bacillus*, *Globicatella*, *Bulleobia* and *Calorana* (Figure 7b). These clusters include genera potentially identified in studies of oral microbiomes. However, the genera *Haemophilus*, *Oribacterium*, *Blastomonas*, *Streptococcus*, *Dialister*, *Sphingomonas*, *Bradyrhizobium*, and were negatively correlated with all the groups mentioned above (Figure 7b).

Functional prediction was performed using Tax4Fun2, a unique tool for predicting and investigating the functional profiles of prokaryotic communities based on sequencing data from the 16S rRNA gene. Metabolic prediction analysis identified 376 pathways in RCs and PPs. Figure 7 illustrates the grouping based on the functional profiles of each sample. The RC cases presented a large number of taxa noted mainly for biofilm formation, resistance to beta-lactams, amino acid biosynthesis and antibiotic biosynthesis. The distribution of the identified pathways is shown in Figure 8a.

Metabolic analyses of the samples were separated into levels related to antibiotic resistance (Figure 8b). Such functional studies allow the prediction of functions related to antibiotic resistance and biosynthesis. PP samples showed a higher number of OTUs related to clavulanic acid biosynthesis, penicillin and cephalosporin biosynthesis, biosynthesis of antibiotics, beta-lactam resistance and arginine and proline metabolism. RCs showed a number of organizational taxonomic units (OTUs) for penicillin biosynthesis, carbapenem biosynthesis, neomycin resistance, clavulanic acid biosynthesis and beta-lactam resistance (Figure 8b).

## 4. Discussion

More than 700 bacterial species have been identified in the oral cavity [25], all of which can lead to contamination of the dental pulp and RCs [26]. Many of these microorganisms are potential agents for infection in the RCs and PPs [2,27,28,29,30,31,32].

Although much is known about the extrinsic factors of endodontic-periodontal lesions combined with relationships between the microorganisms that cause these lesions, little is known about the microbial interrelationship and the diversity of the taxa, which motivated the development of this study to understand the microbiota. Our studies confirmed a large number of works on RC microbial populations, including *Eubacterium*, *Fusobacterium, Peptostreptococcus*, *Porphyromonas*, *Prevotella and Streptococcus* genera [2,27,28,29,30,31,32]. However, other extremely important genera were identified among the most prevalent in RCs, namely, *Enterococcus*, *Parvimonas*, *Stomatobaculum*, *Peptoniphilaceae [G-1]*, *Mogibacterium*, *Olsenella*, *Bacteroidaceae [G-1]*, *Filifactor*, *Oribacterium* and *Pseudomonas*, among others.

The results for PPs are composed of species linked to the genera *Streptococcus*, *Desulfobulbus*, *Enterococcus*, *Parvimonas*, *Saccharibacteria (TM7)*, *Fusobacterium*, *Peptoniphilaceae*, *Actinomyces* and 73 more were detected. Next-generation sequencing studies revealed that the RC and PP microbiome have a rich and common microbial community [2,27,28,29,30,31,32], which is in accordance with our findings.

Although the technology used to assess the microbial community of endodontic-periodontal diseases presents a much greater throughput in relation to traditional [6,33,34,35] cultivation methods for classification regarding gaseous needs, gram-stain, and cell morphology, it was possible to visualize a good part of cultivable and allocated microorganisms in groups, such as gram-positive, gram-negative, aerobic and anaerobic, with emphasis on gram-positive and anaerobic that played a leading role in the RC microbiome. As expected, our bioinformatics analyses of next-generation sequencing data offer greater robustness for detecting microorganisms and also allow us to assess them for their microbiological characteristics, which does not dispense with classical microbiology, but expands the scope of microbial classification [36].

Studies by Gomes et al. [2] revealed a much more diverse microbial population in PPs than in RCs, which corroborates our alpha and beta diversity studies, as beta diversity studies revealed a similar profile between the microbial community of RCs and PPs, which reinforces the fact that there are few variations between the community of RRs and PPs. The calculations of observed OTUs, Shannon and Chao, revealed that PPs have a richer and more diverse microbiota, reinforcing the findings of Gomes et al. [2]. This is because the survival of microorganisms (fungi and bacteria) in the described environment is directly linked and dependent on five factors such as: presence of nutrients, anaerobic environmental conditions, pH value, availability of nutrients which leads to competition, as well as positive interactions with other microorganisms and intrinsic and extrinsic factors of the environment [37,38,39,40,41,42,43]. The factors described above make the environment a determining factor for which microorganisms survive, and some succeed in unfavorable conditions [44].

Many periodontal pathogens are also endodontic pathogens [2,42,44,45]; however, there are a few studies dedicated to massive analysis of next-generation sequencing, such as combinations of annotations and databases related to microorganisms. A small number of articles have focused on the study of the microbiota of RCs and PPs in a holistically and combined manner.

Statistical methods were used to assess the relationship between the RC and PP microbial community, which made it possible to observe a correlation between the endodontic-periodontal microbiota, although a greater number of unique genera for PPs were visible. This difference in the number of genera identified in PPs (highlighted in yellow in Figure 3) is fully related to survival factors as the microbial community in the channel suffers a strong selective pressure driven by the anaerobic environment, nutrient availability [46], and competition and interaction between microbial species [38], clearly explaining the higher number of genera in PPs.

The endodontic-periodontal microbiota consists of a community of similar microorganisms, as can be seen in the Venn diagrams, where all the phyla and genera identified in RCs are part of the PP microbiota, corroborating the findings of Rupf et al. [31] who identified putative periodontal pathogens in RC infections. Kipioti et al. [28], Kobayashi et al. [29], and Kurihara et al. [47] reported that advanced periodontitis can be a gateway for microorganisms, which leads to severe infections in RCs. Although this communication and dispersal of microorganisms is extremely consolidated, it is still quite difficult for the portion of the microbiota that causes the problem and those simply residing because of environmental factors [48]. The literary evidence described above is fully reinforced by the clusters, which group cases of RCs and PPs, the same can be seen in the hierarchical groupings that reveal clusters of PP and RC cases.

An analysis of co-relation and co-occurrence allowed us to group the microorganisms into conglomerates, that is, groups resulting from positive and negative interactions. *Helicobacter*, *Sulfurospirillum*, *Atopobium*, *Olsenella*, *Bosea*, *Kocuria*, *Hyphomicrobium*, *Rhodobacter*, *Paracoccus*, *Rhizobium*, *Bacillus*, *Globicatella*, *Bulleobia* and *Calorana* were the genera with positive interactions between RCs and PPs, which reinforces the data from Socransky et al [49], as the microorganisms in the conglomerates resulting from positive interactions are not part of the groups of the purple, yellow, green, or orange complexes, concluding that there really are microbial groups that survive in isolation in each compartment, including the RC and PPs.

The development of metagenomics associated with next-generation sequencing has led to the development of bioinformatics tools that allow the inference of the functional profile (proteins, resistance genes, and disease markers, among others) of a microbial community based on the research of marker genes in one or more samples [50]. The metabolic studies of endodontic-periodontal diseases conducted in this research revealed a potential with regards to systemic problems caused by the microbial community at both sites, such as heart failure, rheumatology and myocarditis, among others, which reinforces the hypothesis of numerous studies confirming the presence of bacteria from infected RCs in the bloodstream, due to the occurrence of bacteremia, as a determining factor for systemic complications [51,52].

Antibiotic therapy was a major advance without significant decline in potentially fatalities, initiating a new era of infectious disease therapy [53,54]. Many years after this pharmaceutical evolution with antibiotic therapy, it brought severe causes with regard to evolutionary responses to the selective pressure exerted by antibiotics, resulting in microbial species resistant to virtually all known drugs (antibiotics),which reinforces the results obtained in this research since many markers for antibiotic resistance genes were found [53,55,56], reinforcing the literature indicating that bacteremia may be associated with bacterial endotoxins from infected canals, presenting risks for systemic complications. Furthermore, such risks can be minimized when appropriate therapeutic measures are applied, such as careful handling of infected canals and prophylactic antibiotic therapy for high-risk patients [52].

## 5. Conclusions

The RC microbiota was less complex and diverse than PP. There were interactions between the microbial communities. Shared species can signalize communication between the endodontic and periodontal microbiomes. We also emphasize that microorganisms related to endodontic-periodontal diseases have a strong correlation with systemic diseases.

## Figures and Tables

**Figure 1 microorganisms-09-01925-f001:**
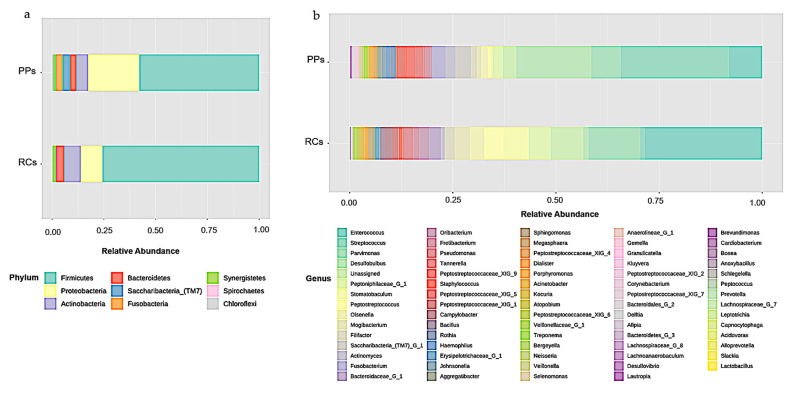
Relative abundance of (**a**) phyla (**b**) bacterial genera in root canals (RCs) and periodontal pockets (PPs).

**Figure 2 microorganisms-09-01925-f002:**
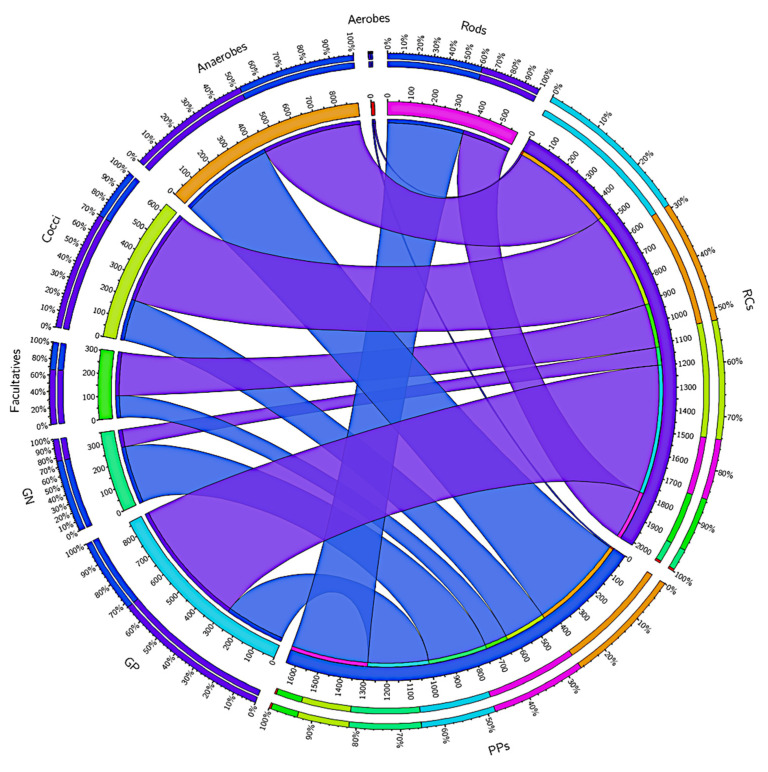
Distribution of the reads in the root canals (RCs) and periodontal pockets (PPs) accord-ing to phenotypic characteristics of the taxa present at the both sites, such as Gram-positives, Gram-negatives, rods, cocci, aerobes, facultatives and anaerobes. Data visualization was done through Circos software. The thickness of the bars, as the outer ring represents the percentage of readings in each study condition.3.2. Analysis of alpha and beta diversity in the RCs (represented in blue) and PPs (represented in purple) microbiomes.

**Figure 3 microorganisms-09-01925-f003:**
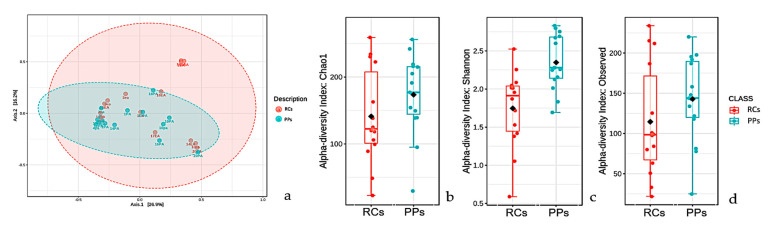
Beta diversity analyzes performed using principal coordinates (PCoA) based on the Jensen-Shannon distances between bacterial communities in the studied conditions (**a**) root canals (RCs) and periodontal pockets (PPs) (ANOSIM, R = 0.3066; *p*-value < 0.001).(**b**) Alpha diversity at the observed level OTUs of RCs (red, *n* = 12) and PPs (cyan, *n* = 12) calculated considering the number of observed OTUs (Kruskal-Wallis test, value of *p* < 0.05). (**c**) Alpha diversity at the level of OTUs in the RCs (red, *n* = 12) and PPS (cyan, *n* = 12) calculated using the Shannon index (Kruskal-Wallis test, *p* value < 0.05). (**d**) Alpha diversity at the level of OTUs in the RCs (red, *n* = 12) and PPs (cyan, *n* = 12) calculated considering Chao 1 indices. For each boxplot presented, the graph grids represent the median and the mean, respectively. The lower and upper limits of each box indicate the first and third quartiles, respectively. The lines represent the lowest and highest values within the 1.5 interquartile range (IQR).

**Figure 4 microorganisms-09-01925-f004:**
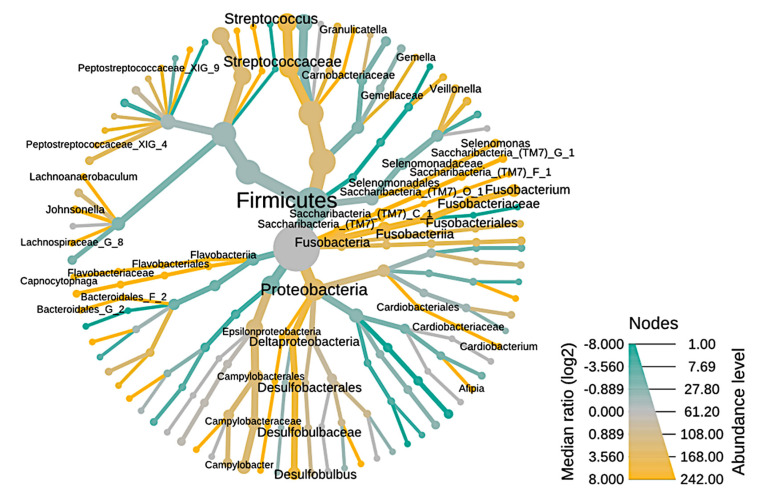
Heat tree illustrating the general taxonomy of the bacterial community through root canals in relation to periodontal pockets. The nodes represented by the green color are the most abundant (enriched) in RCs, the yellow ones are the most prevalent in PPs. The nodes represented by the gray color are present in the same proportion in both study conditions.

**Figure 5 microorganisms-09-01925-f005:**
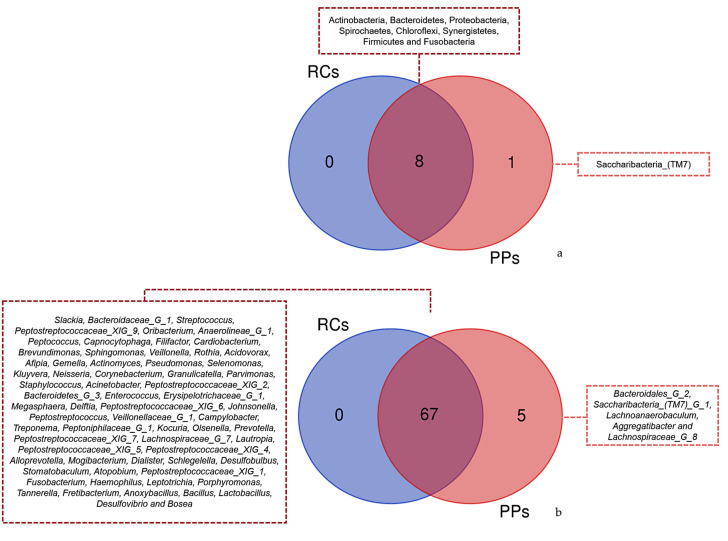
Venn diagram showing overlap of the identified reads for (**a**) Phyla between the different samples root canals (RCs) and periodontal pockets (PPs). Overlapping of (**b**) Genus with the highest number of readings identified between different samples Root Canals (RCs) and Periodontal Pockets (PPs).

**Figure 6 microorganisms-09-01925-f006:**
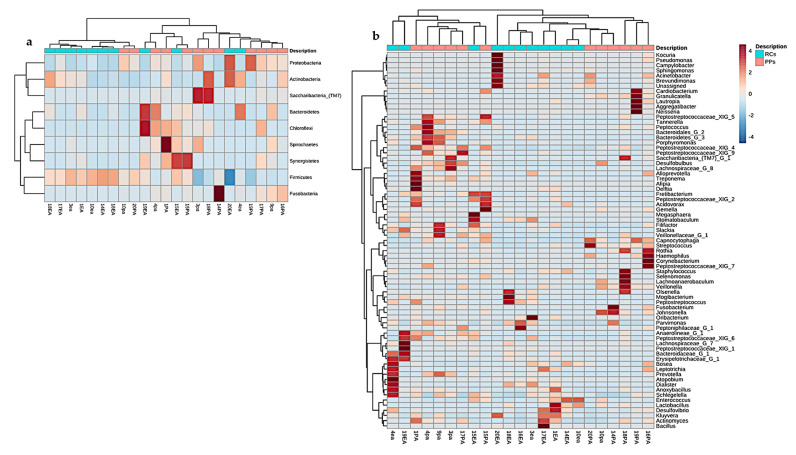
Heatmap containing similarities of the (**a**) phyla (**b**) genera in root canals and periodontal pockets. Red bars represent categories of phyla with the highest prevalence in the 12 samples of each site analyzed with a diversity pipeline.

**Figure 7 microorganisms-09-01925-f007:**
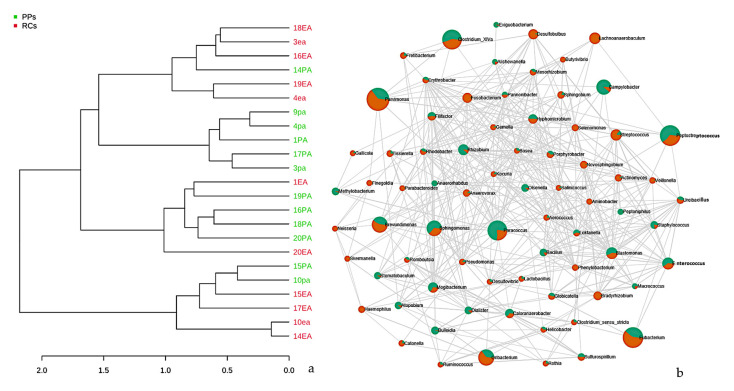
(**a**) Dendrogram showing the similarities between the samples of RCs and PPs. The dendrogram was created using the Jaccard index as a distance measure and Ward’s clustering algorithm considering a 1000-repetition bootstrap. (**b**) Correlation and co-occurrence analysis of cases of root canals (green) and periodontal pockets (orange) genera. The interactions (connections) represent the relationship (Spearman’s ρ > 0.6) and the significance (*p* < 0.01). Circles represent the relative abundance of each condition of study; the thickness of each connection between two nodes is proportional to the value of Spearman’s correlation coefficients.3.4. Metabolic prediction in RCs and PPs.

**Figure 8 microorganisms-09-01925-f008:**
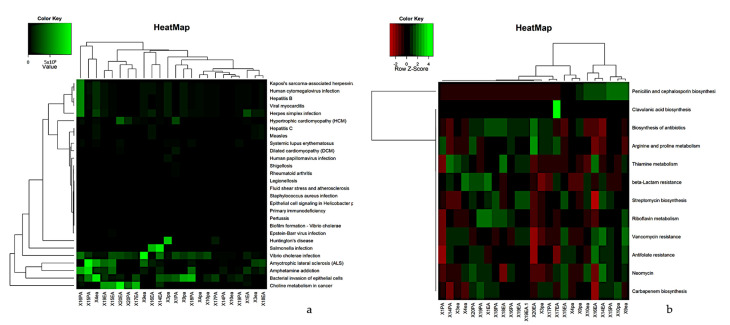
A hierarchical cluster heat map that shows the tax4Fun (**a**) functional prediction in metagenomes of the micro-bial community (**b**) Functional prediction study of health-related genes of 12 samples of RCs and PPs. The metabolic pathways (resulting from the prediction) are in the rows and the RCs and PPs samples are in the columns. Columns are centered on the mean, with relative abundance represented by colors (black and red, lowest abundance; green, highest abundance) as indicated in the legend.

## Data Availability

The metagenomic data (RCs and PPs) were deposited in the NCBI GenBank database under accession number QNUO01000000. Data are deposited at https://www.ncbi.nlm.nih.gov/genbank/ (accessed on 22 August 2021) with download available using the access number described above.

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
