# Peer review of "Interrelationship between the Microbial Communities of the Root Canals and Periodontal Pockets in Combined Endodontic-Periodontal Diseases"

_microorganisms, 2021, doi:10.3390/microorganisms9091925_

Round 1
Reviewer 1 Report
In clinical practice, the prognosis for teeth with endodontic and periodontal lesions is unfavorable and treatment is often difficult. The cause may be related to bacteria in the root canal or periodontal pocket, but there are few reports on these bacteria. In this study, the authors provide new information regarding next-Generation Sequencing analysis of root canal microbiota associated with a severe endodontic-periodontal lesion.
This topic is important and will be of great interest to readers of the journal. Thank you for the answer and adjusting point to some of my questions. There is no point in particular to worry about the answer and correction to questions. I’m looking forward to expecting development of the further research on this field.
Author Response
In clinical practice, the prognosis for teeth with endodontic and periodontal lesions is unfavorable and treatment is often difficult. The cause may be related to bacteria in the root canal or periodontal pocket, but there are few reports on these bacteria. In this study, the authors provide new information regarding next-Generation Sequencing analysis of root canal microbiota associated with a severe endodontic-periodontal lesion.
This topic is important and will be of great interest to readers of the journal. Thank you for the answer and adjusting point to some of my questions. There is no point in particular to worry about the answer and correction to questions. I'm looking forward to expecting development of the further research on this field.
Response:
Dear Reviewer 1,
We thank you for your availability to evaluate this paper. Your considerations are of great value for this work and for the next ones that will come in the same field.
We have submitted again the revised manuscript to the English Revison. Please find attached the English Correction Certificate. Please find also attached the manuscript with the necessary revisons (marked in gray).
Yours sincerely,
The authors

Reviewer 2 Report
The manuscript describes the analysis with next-generation sequencing of samples from 12 patients with combined endodontic-periodontal diseases. The manuscript was well-written, and suitable for the publication in the Microorganisms.
[Suggestions]
L. 325-326: "(highlighted in yellow in Figure 3)"
The referee is unable to find it in the Figure 3; thus, maybe in the Figure 1?
Author Response
Reviewer 2:
The manuscript describes the analysis with next-generation sequencing of samples from 12 patients with combined endodontic-periodontal diseases. The manuscript was well-written, and suitable for the publication in the Microorganisms.
[Suggestions]
- 325-326: "(highlighted in yellow in Figure 3)"
The referee is unable to find it in the Figure 3; thus, maybe in the Figure 1?
Response:
Dear Reviewer 2,
Thank you for your comments. In order to make Figure 3 clearer and more explanatory, we added the following sentence:
(…) The abundance of taxa corroborates the data described in Figure 2, where Cardiobacterium, Streptococcus, Campylobacter, Pseudomomas, among others are represented by the lines and nodes in the green color, which indicates greater abundance in RCs (Figure 3), whereas the PP microbiota has high prevalence of Fusobacterium, Streptococcus, Capnocytophaga among others visually represented in yellow lines and nodes. Figure 3 clearly represents that there are more genera identified in PPs than in RCs, as there are a larger number of yellow nodes than green nodes.(…).

Reviewer 3 Report
The manuscript “Interrelationship between the microbial communities of the root canals and periodontal pockets in combined endodontic-periodontal diseases” presents a study evaluating the microbial composition of infected periodontal pockets and root canals to further assess the microbiota and verify the possibility of cross-infection between these sites.
In the manuscript, the question is original and well defined and the results provide an advance in current knowledge; the results are interpreted appropriately; all conclusions are justified and supported by the results; the article is written in an appropriate way; the data and analyses are presented appropriately; the study is correctly designed and technically sound; the analyses are performed with the highest technical standards; the methods, tools, software, and reagents are described with sufficient details to allow another researcher to reproduce the results; the conclusions are interesting for the readership of the Journal and the paper presumably will attract a wide readership; there is an overall benefit to publishing this work; the English language is appropriate and understandable.
The only comment that I would like to suggest to the authors is the following:
There are some (quite a few) typos. Please double check all the text and correct the typos. (es. Lines 55-56; 72, 84-85, etc.)
Furthermore paragraph 2.2 should be reworded and the meaning of CRs (line 118) explained.
For the reasons listed above, my final recommendation is to accept after minor revisions the manuscript.
Best regards
Author Response
The manuscript “Interrelationship between the microbial communities of the root canals and periodontal pockets in combined endodontic-periodontal diseases” presents a study evaluating the microbial composition of infected periodontal pockets and root canals to further assess the microbiota and verify the possibility of cross-infection between these sites.
In the manuscript, the question is original and well defined and the results provide an advance in current knowledge; the results are interpreted appropriately; all conclusions are justified and supported by the results; the article is written in an appropriate way; the data and analyses are presented appropriately; the study is correctly designed and technically sound; the analyses are performed with the highest technical standards; the methods, tools, software, and reagents are described with sufficient details to allow another researcher to reproduce the results; the conclusions are interesting for the readership of the Journal and the paper presumably will attract a wide readership; there is an overall benefit to publishing this work; the English language is appropriate and understandable.
The only comment that I would like to suggest to the authors is the following:
There are some (quite a few) typos. Please double check all the text and correct the typos. (es. Lines 55-56; 72, 84-85, etc.)
Furthermore paragraph 2.2 should be reworded and the meaning of CRs (line 118) explained.
Response:
Dear Reviewer,
Thank you for your comments about this manuscript. We carried out an extensive proofreading in the text to check for possible typos.
Paragraph 2.2 has been rewritten as seen below:
(..) Periodontal pocket and root canal samples were collected before starting both treatments. The samples were collected following the protocol established by Gomes et al. (2015). Three consecutive sterile paper points were kept in place for 60s, and then added into a sterile tube containing 1 mL of VMGA III transport medium, followed by freezing at -70 °C until processing.
